# Banana Pseudo-Stem Increases the Water-Holding Capacity of Minced Pork Batter and the Oxidative Stability of Pork Patties

**DOI:** 10.3390/foods10092173

**Published:** 2021-09-13

**Authors:** Diego E. Carballo, Irma Caro, Cristina Gallego, Ana Rebeca González, Francisco Javier Giráldez, Sonia Andrés, Javier Mateo

**Affiliations:** 1Department of Hygiene and Food Technology, Faculty of Veterinary Medicine, University of León, 24071 León, Spain; diegocarballo2@hotmail.com (D.E.C.); cgallg00@estudiantes.unileon.es (C.G.); rebbeka1196@gmail.com (A.R.G.); 2Department of Nutrition and Food Science, Faculty of Medicine, University of Valladolid, 47005 Valladolid, Spain; irma.caro@uva.es; 3Comercializadora GONAC SA de CV, Camino Nacional No. 7. Ciudad Industrial Xicohtencati II, Huamantla 90500, Mexico; 4Instituto de Ganadería de Montaña, CSIC-Universidad de León, Finca Marzanas s/n, Grulleros, 24346 León, Spain; j.giraldez@eae.csic.es (F.J.G.); sonia.andres@eae.csic.es (S.A.)

**Keywords:** discolouration, lipid autoxidation, natural antioxidant, functional food, cooking yield, patty

## Abstract

Banana pseudo-stem (BPS), which is rich in fibre and polyphenols, is a potential functional ingredient for the food industry. In this study, BPS was added at concentrations of 1.5, 3.0, and 4.5 g/kg to a minced pork batter to evaluate its performance as a filler and to pork burger patties to evaluate its performance as a natural antioxidant. The effects of BPS were compared with those of carrageenan and ascorbate, which are a conventional binder and antioxidant, respectively. The performance of BPS was similar to that of carrageenan in terms of the cooking yield and texture of the cooked batter. BPS reduced the brightness of fresh patties and appeared to reduce oxidative discolouration during the frozen storage of raw patties. Moreover, BPS reduced the levels of thiobarbituric acid reactive substances (TBARS) during the refrigerated and frozen storage of cooked patties. A greater decrease in TBARS formation was observed with 4.5 g BPS/kg compared with 0.5 g sodium ascorbate/kg during refrigerated storage. In contrast to ascorbate, BPS promoted the presence of lipid-derived volatile compounds induced by thermal breakdown in the headspace of cooked patties. Nonetheless, this effect was reduced as the amount of BPS in the patties increased. In cooked minced meat products, BPS could increase cooking yields and lipid oxidative stability during storage and might result in a more intense flavour.

## 1. Introduction

Meat products are highly susceptible to oxidative changes that limit their shelf life, such as discolouration during the storage of raw meat products or the development of rancid or warmed-over off-flavours during the storage of cooked meat products [1,2,3]. The use of antioxidants is a well-known and frequently used strategy to control and minimise these oxidative changes. In this context, the meat industry has shown increasing interest in the use of natural antioxidant-rich sources or their extracts, mainly of vegetal origin, which is partly due to a growing consumer preference for additive-free foods [4,5,6,7]. Among the multiple sources of natural antioxidants, there is a range of agroindustry by-products with high contents of phenolics and other active ingredients [8,9,10]. One that is arousing growing interest is banana pseudo-stem (BPS) [11].

Banana, *Musa acuminata*, is widely produced in humid tropical and subtropical regions in the world. The production of bananas generates a large number of by-products, and among them is BPS. BPS can be incinerated or left in the plantation soil as a waste residue, or it can be used as bio-fertiliser or for manufacturing purposes in a variety of industrial areas, from animal feed to the cultivation of edible mushrooms, paper production, or application as a raw material for textiles or cement reinforcement [11,12,13]. According to Mohapatra, Mishra, and Sutar [14], in India, BPS uses include food preparation, where the pith of BPS is used as an ingredient in flours, jams, drinks, and confectionery products.

As a result of the health-promoting and technological functional properties attributed to BPS, its use as a food supplement or functional food ingredient has been suggested. According to Bhaskar, Mahadevamma, Chilkunda, and Salimath [15], BPS, due to its high content of dietary fibre and polyphenols, could promote beneficial health effects. Its high starch and fibre contents provide the product with high water-holding capacity and thickening and gelling properties, and as a polyphenol-rich material, it can exert antimicrobial and antioxidant effects in food [11,16].

Saravanan and Aradhya [17] stated that BPS is a potential source of natural antioxidants for the food and nutraceutical industry. They reported that BPS contains high levels of polyphenols, such as protocatechuic acid, gallic acid, caffeic acid, ferulic acid, and tannic acid, among others. Furthermore, the scavenging effect provided by an acetone extract of BPS at 100 μg/mL on the 1,1-diphenyl-2-picrylhydrazyl (DPPH) radical was between 41% and 70%, depending on the cultivar, while the scavenging effect on butylated hydroxytoluene (BHT) under the same conditions was 93%.

Recent studies have investigated the potential of BPS as a natural antioxidant ingredient in foods. Ho, Abdul Aziz, and Azahari [18] used BPS (up to 10%) mixed with wheat flour to prepare bread. Their results showed that BPS increased the amounts of ash and dietary fibre and provided antioxidant capacity without altering panellist acceptability ratings. Anusuya, Gomathi, Tharani, and Murugesan [19] investigated the effect of BPS in sunflower oil and suggested that BPS extract, rich in polyphenols, could serve as a substitute for synthetic antioxidants in edible oils. However, no studies have reported the use of BPS in the formulation of meat products. In this context, the novelty of this research lies in evaluating the potential of BPS as a functional ingredient in the meat industry. In particular, the filler effect of BPS in cooked meat batter was evaluated and compared with that of carrageenan (CAR), and its antioxidant effect in burger patties was analysed and compared with that of ascorbate.

## 2. Materials and Methods

### 2.1. Ingredients

Fresh boneless pork loins were purchased from a local meat retailer, and food-grade carrageenan (Danagel, FMC ByoPolimer, Brussels, Belgium), sodium ascorbate, sodium chloride, and spices were obtained from local suppliers. Dried powdered BPS was provided by Mfood Alimento Natural (León, Spain). This was obtained from the Canary Islands banana plantations by cutting the pseudo-stem longitudinally, which was then ripened and sun-dried for three weeks, cut, dried with hot air (65 °C), and ground into a powder. The powdered BPS composition characteristics were analysed in duplicate, and the results are shown in Table 1.

### 2.2. Experimental Plan

This research was composed of two experiments. The first experiment evaluated the performance of BPS as a filler in minced lean-pork batter. BPS was compared with a control (CON) batter (with no filler) and batter with CAR. A total of seven batter mixtures were prepared. Three of them contained BPS at concentrations of 1.5, 3.0, and 4.5 g/kg batter, designated as low (LBPS), medium (MBPS), and high (HBPS) levels, respectively. Another was made without fillers (CON), and the other three contained CAR at the same levels as those used for BPS (LCAR, MCAR, and HCAR). The quality traits analysed were the pH of the raw pork batter, cooking losses, and texture profile analysis of the cooked pork batter. The amounts of BPS and CAR used were within the range reported for meat products for CAR [24,25]. Moreover, in a preliminary consumer study (*n* = 117), we found that the use of 4.5 g BPS/kg pork batter did not negatively affect their sensory acceptance as compared with a patty with 4.5 g CAR/kg (the mode value obtained in the sensory test for the BPS patty was 7, and that for the CAR patty was 6, *p* = 0.189, Mann–Whitney test; unpublished data).

In the second experiment, five types of lean-pork burger patties were prepared. The effect of using BPS (at the above-mentioned LBPS, MBPS, and HBPS levels) on the oxidative stability of the patties was evaluated and compared with a CON patty (with no antioxidants) and a patty containing 0.5 g/kg sodium ascorbate (ASC), which is an antioxidant additive commonly used in the meat industry [24]. Colour stability was determined in fresh patties, and the patty colour was compared before and after 3 months of frozen storage. Lipid oxidative stability was determined in cooked patties by evaluating the formation of thiobarbituric acid reactive substances (TBARS) during refrigerated (2 days) and frozen (3 months) storage. Finally, headspace volatile components in recently cooked patties were analysed to assess lipid thermal stability to oxidative breakdown.

The pork batter and patties were prepared in the Food Processing Hall of the Veterinary Faculty at the University of León, León (Spain). Raw batter and burger patties were prepared in quintuplicate (five batches or independent replications). Each batch consisted of the following experimental levels: seven types of raw batter for the first experiment and five burger patties for the second.

### 2.3. Experiment 1: Preparation of Lean-Pork Patty Batter

Lean-pork batter samples were prepared in quintuplicate, i.e., on five different days. Patty batter samples were prepared from a 2 kg mix, made with 78.5% minced pork loin (butcher’s mincer, 5 mm 15 mm diameter sieve), 20% water, and 1.5% common salt, which were mixed manually for 2 min. Then, seven 250 g portions were taken from the whole mix and assigned to the corresponding experimental treatments: CON, control (with no added ingredients), LBPS, MBPS, HBPS, LCAR, MCAR, and HCAR (with the addition of the above-mentioned amounts of BPS or carrageenan). Then, each portion with the corresponding ingredient was mixed manually for 2 min. Then, minced lean-pork batter samples (35 ± 1 g) were carefully placed in quadruplicate into 50 mL Falcon tubes, and the then, tubes with the samples were stored in the refrigerator overnight before cooking and analysis.

### 2.4. Experiment 1: Evaluation of Water-Holding Capacity and Texture of Cooked Pork Batter

The pork batter pH was measured in duplicate using a puncture electrode (CRISON GLP22, Alella, Barcelona, Spain). Cooking losses were determined following the procedure described by Lin and Mei [26], with some modifications. Briefly, the pork batter samples placed in the tubes were cooked at 80 °C for 30 min in a water bath (Tectron 3473100, Barcelona, Spain). After cooking, the tubes were cooled in an ice bath for 10 min, and the cooked meat batter samples were removed from the tubes and then carefully drained and weighed. In addition, following the methods described by Townsend, Witnauer, Riloff, and Swift [27], the drained juice was poured into a flask, weighed, dried at 100 ± 2 °C for 4 h, and weighed again in order to estimate the solids content in the juice. Then, the cooked batter samples were cut to obtain three 1 cm cubes per sample, which were used for texture profile analysis (TPA). Each meat cube was compressed (80% strain value) in two repeated cycles (5 s between each compression) using a texture analyser (TA.XT2i, Stable Micro Systems, Surrey, England) equipped with a 5 cm diameter cylindrical probe operating at a test speed of 0.5 m/s. The Texture Expert v1.20 software was used to obtain the hardness, springiness, cohesiveness, and chewiness parameters.

### 2.5. Experiment 2: Preparation of Lean-Pork Patties

A total of 3 kg of patty batter was prepared in quintuplicate (on five different days) with 78.5% minced lean (5 mm diameter sieve) pork loin, 20% water, 1.5% common salt, and 1 g/kg pepper. All of the ingredients were vigorously mixed manually (2 min), and the resulting batter was divided into five 0.5 kg portions. One portion was used for CON patties with no further ingredient addition; amounts of 1.5, 3.0, and 4.5 of BPS/kg were added to three portions (LBPS, MBPS, and HBPS portions, respectively), and 0.5 g of sodium ascorbate/kg was added to the other portion (ASC). The batter was vigorously mixed manually for another 2 min, and four 100 g patties were formed from each one. Two patties were used for colour stability analysis, and the other two patties were used to analyse the lipid oxidative stability of cooked patties and the headspace volatile composition.

### 2.6. Experiment 2: Evaluation of the Colour and Lipid Oxidative Stability of Patties

Colour was measured immediately, in triplicate, directly on the upper surface of the fresh patties using a Konica Minolta CM-700d colourimeter (Osaka, Japan) operating with D65 illuminant, a visual angle of 10°, and an 11 mm aperture for illumination and 8 mm for measurement, in SCI mode. Then, the patties were wrapped in polyvinyl chloride cling film and stored for 90 days at −18 °C, thawed for 48 h at 4 °C, unwrapped, and tempered for 2 h at room temperature (22 °C), and the colour was measured as described before.

For the evaluation of lipid oxidative stability, patties were first cooked in a convection oven for 150 °C until reaching a core temperature of 70 ± 2 °C (for approximately 20 min), and then, each patty was divided into two halves. Then, one of the halves was divided into three equal portions: the first was used for immediate analysis of TBARS (TBARS in recently cooked patties), and the second and the third were wrapped individually in polyvinyl chloride cling film: one was stored at 4 ± 2 °C in darkness for two days, and the other was stored at −18 °C for three months before TBARS analysis. This analysis was carried out in duplicate following the procedure described by Nam and Ahn [28]. The frozen patty portion was thawed at 4 °C for one day before analysis.

The remaining halves were used for the analysis of volatile compounds, which was carried out in duplicate by gas chromatography coupled to mass spectrometry following the procedure described by Carballo, Caro, Andrés, Giráldez, and Mateo [29] with modification of the extraction temperature. Briefly, 2 g of homogenised cooked patty sample was placed in 15 mL screw-cap vials (Agilent Technologies, Santa Clara, CA, USA), which were then intermittently agitated (500 rpm, 5 s on, 2 s off) at 110 °C for 40 min in the autosampler. CG 7890A equipment (Agilent Technologies, Santa Clara, FL, USA) was used. The headspace injection volume was 1 mL, and the injector was operated in splitless mode at 260 °C. The syringe needle temperature was 120 °C, and the filling and injection speeds were 50 and 250 μL/s, respectively. A DB-5MS column (60 m × 0.25 mm ID × 0.25 μm film thickness; J&W Scientific, Folsom, CA, USA) was used for separation, and helium at a flow rate of 1.5 mL/min was the carrier gas. The oven conditions were as follows: 38 °C held for 1 min as the initial temperature, which was increased to 50 °C at 10 °C/min, then to 146 °C at 4 °C/min, then to 200 °C at 20 °C/min, and finally to 250 °C at 50 °C/min, which was maintained for 11 min. The transfer-line temperature was 260 °C. For detection, a simple quadrupole mass spectrometer (MSD 5975C, Agilent Technologies) was operated with an electron source temperature of 240 °C, a quadrupole temperature of 190 °C, electron energy and emission currents of 70 eV and 35 μA, respectively, and a scanner range from 40 to 230 m/z. Tentative identification of volatile compounds was performed by comparing the spectra of the detected volatile compounds with those in the NIST/EPA/NIH-98 Mass Spectral Database and comparing their linear retention indices, which were calculated from the retention times of a series of n-alkanes (Hydrocarbons/C5–C20; Sigma-Aldrich, St. Louis, MO, USA), with those found in the literature [30,31,32].

### 2.7. Statistical Analysis

The statistical analysis was performed using SPSS Statistics software (version 26; IBM, Somers, NY, USA). Data were analysed using the general linear model analysis of variance, with treatment (ingredients used) as the fixed factor and batch as a random factor. Comparison between storage times or conditions within treatments were analysed using Student’s *t*-test for colour data and ANOVA for TBARS data. In this case, storage condition (non-stored, refrigerated storage, and frozen storage) was used as a fixed factor and batch was used as a random factor. ANOVA was followed by the Tukey test to assess differences between treatments, storage times, or conditions. Pearson correlations between selected variables were also carried out.

## 3. Results and Discussion

### 3.1. Experiment 1: Effect of BPS on the Water-Holding Capacity and Texture of Pork Batter

Table 2 shows the effects of adding BPS or CAR at concentrations up to 4.5 g/kg to minced lean-pork batter on yields and texture. The batter pH increased with the increase in the BPS amount, although the change was lower than 0.1 pH units. In contrast, CAR decreased the batter pH, with these changes also lower than 0.1, and the extent of the decrease remained unchanged regardless of CAR content. CAR is considered a dietary fibre and is frequently used as a meat binder. Both ingredients reduced the cooking losses to similar degrees, and the reductions were proportional to the amount of the ingredient used. The ingredients also tended to reduce hardness, which was significantly higher in CON batter than in the others, with the exception of LBPS (*p* < 0.05). The solids in the fluid lost during cooking and the elasticity, cohesiveness, and chewiness of cooked batter were not affected by the treatment (*p* > 0.05).

The effect of CAR on cooking yield (water retention) in meat products has been demonstrated in previous studies [33,34]. CAR in cooked minced meat products also contributes to gel formation [25], and thus, it can increase the hardness of the meat batter after cooking. However, according to Ayadi et al. [33], and in agreement with this study, the increase was noticeable at concentrations of >5 g/kg, which are higher than those used in this study. BPS, a fibre-rich ingredient, has a high water-holding capacity compared with other dietary fibres, such as oat, rice bran, and even soy flour [16]. Overall, the results of this study show that CAR could be replaced by BPS at levels up to 4.5 g/kg in cooked meat products without affecting the technological quality traits studied. Both ingredients improved the cooking yield to similar extents.

### 3.2. Experiment 2: The Effect of BPS on Colour and Lipid Oxidative Stability of Burger Patties

Table 3 shows the effect of using ascorbate and BPS on the colour values (L*, a*, b*, chroma, and hue angle) of recently prepared and frozen-stored and thawed patties. The addition of BPS resulted in a decreased L*, indicating a darkening effect, in the fresh patties (*p* < 0.001); however, no change in L* due to BPS was observed in the frozen-stored patties. The other colour values (a*, b*, chroma, and hue angle) were not significantly affected in either fresh or frozen/thawed patties. The effect of adding fibre-rich ingredients to patties on their colour depends on the amount and the colour of the ingredient used, which is beige-brown in the case of BPS. Other studies have also found that rice bran or hazelnut pellicle addition reduces brightness (L*) in meat batter and burgers [35].

In general, patties tend to show discolouration during frozen storage, which, together with rancidity due to lipid oxidation, limits their shelf life. Discolouration due to frozen storage has been attributed to myoglobin autooxidation [35,36,37]. To retard the formation of metmyoglobin in frozen-stored patties, the use of antioxidants has been suggested [38]. In the present study, discolouration was evidenced in both CON and ASC patties by significant decreases in a* and chroma and an increase in hue angle during frozen storage [39].

In addition, AMSA [39] proposed the use of two visible spectral calculations to evaluate changes in the redness of raw meat due to variations in either oxymyoglobin or deoxymyoglobin contents and subsequent metmyoglobin formation during storage: the wavelength ratios of reflectance 630 nm/580 nm (R_λ630_/R_λ580_, with higher ratios indicating more redness) and 572 nm/525 nm (R_λ572_/R_λ525_, with higher values indicating more discolouration). Table 3 shows the values of these ratios in fresh patties, as calculated from their colour spectra. The use of BPS in fresh patties led to a decrease in R_λ630_/R_λ580_ and a subsequent increase in R_λ572_/R_λ525_, and the changes were directly dependent on the BPS amount used. These changes could be the result of a direct effect of BPS on the light reflectance of patties rather than a particular effect of BPS on the proportions of the three myoglobin chemical forms. After frozen storage, the R_λ630_/R_λ580_ and R_λ572_/R_λ525_ values, which significantly differed from those of the fresh patties, were similar between groups.

Table 3 also shows the changes in R_λ630_/R_λ580_ and R_λ572_/R_λ525_ for each group of patties due to frozen storage. These were calculated by subtracting the frozen patties’ reflectance ratios from those of the fresh patties. An R_λ630_/R_λ580_ decrease and an R_λ572_/R_λ525_ increase were found, which were significantly diminished by the presence of BPS. This finding suggests [39] a lower metmyoglobin formation rate, i.e., lower discolouration during frozen storage in the BPS patties, and thus a protective effect of BPS on myoglobin stability to oxidation.

The values of TBARS in recently cooked patties, in cooked and then refrigerated patties (for 48 h), and in frozen-stored raw patties (for three months) are shown in Table 4. Lipid oxidation occurred during both refrigerated and frozen storage of cooked and raw patties, respectively. Oxidation was more pronounced after two days of refrigerated storage than after three months of frozen storage, which is explained by the exceptionally high lipid oxidation rate in meat substrates after heat treatment [40].

The antioxidant effect of BPS was clearly evidenced by changes in the TBARS contents in the cooked patties; i.e., TBARS contents were significantly lower in BPS-containing patties than in CON cooked patties before storage and after refrigerated and frozen storage. The antioxidant effect of BPS (the TBARS reduction) was directly related to the amount used. For the cooked patties stored in the refrigerator, a better, similar, and worse performance was found with the use of 4.5, 3.0, and 1.5 g BPS/kg, respectively, compared with the use of 0.5 g ascorbate/kg. For the frozen-stored patties, only BPS at a concentration of 4.5 g/kg led to a significant reduction in TBARS as compared with controls, and ascorbate was always more efficient than BPS in retarding TBARS formation.

The antioxidant effect of BPS has been reported in previous studies [16,18]. When used as an ingredient in different foods, it was found that the addition of BPS to wheat flour significantly improved the antioxidant capacity of bread [18]. However, no studies have been carried out on BPS in meat products, so there are no literature values to compare with the present results.

### 3.3. Experiment 2: Headspace Volatile Composition in Cooked Patties

A total of 47 volatile compounds released by the cooked patties to the headspace after 40 min of incubation at 110 °C were identified in this study: 10 aliphatic aldehydes, 2 ketones, 4 alcohols, 6 furans, 8 hydrocarbons, 13 terpene compounds, 2 benzene compounds, and 2 sulphur compounds. For brevity, Table 5 includes the data for compounds showing concentrations higher than 4 ng equivalent of hexane/mL of headspace in any of the treatments. The aldehydes, ketones, alcohols, furans, and hydrocarbons detected are volatile compounds originating from the thermal breakdown of lipids [41,42,43]. Thus, they would have been formed mainly during the heating of the patties, and their levels would be related to the oxidative stability of lipids to heat treatment.

Most of the lipid-derived compounds and the total sum of volatiles were significantly affected by antioxidant treatment. On the one hand, the use of ascorbate resulted in the lowest levels of these volatiles in the patty headspace. This suggests that ascorbate has a protective effect on lipids against their thermal breakdown. In agreement with these results, other studies have found that the use of antioxidants, such as ascorbate, rosemary extract, or astaxanthin in cooked patties, minced meat, or meat models, exerted a suppressive effect on the formation of lipid-derived volatile compounds as compared with controls [29,44].

In contrast, the use of BPS, despite its ability to promote lipid oxidative stability in the patties during storage (Table 3), resulted in a higher concentration of lipid-derived volatile compounds in the patty headspace (presumably originated by thermal breakdown) compared with CON. The effect of BPS on the concentration of these volatile compounds was not proportional to the amount used. On the contrary, the increasing effect of BPS tended to weaken as the amount of BPS was increased (from 1.5 to 4.5 g/kg patty). This trend suggests a balance between the protective and pro-oxidative effects of BPS compounds on thermal lipid breakdown.

Two types of mechanisms might be involved in the above-mentioned complex effect of BPS on the presence of lipid-derived volatile compounds in the patty headspace. One of them could be associated with changes in the patty matrix structure due to the presence of BPS, which would affect the release of these flavour compounds to the headspace. The other could be related to chemical interactions between specific BPS compounds and the meat lipids, which would affect the lipid thermal breakdown rate during patty heating.

The changes in the patty matrix structure might be attributed to the slight changes in pH produced by BPS (Table 2) and also to the interaction of minerals from BPS, especially those forming soluble salts, such as CaCl_2_ or MgCl_2_, with meat proteins. Changes in the protein matrix affect their ability to bind lipid-derived volatile compounds [45], which, in this case, would be reduced. On the other hand, the transition metals (e.g., iron) incorporated in the patties by BPS might have exerted a catalysing effect on lipid thermal breakdown reactions [46] during patty heating, which would have been more potent than the protective effect of BPS antioxidant compounds. Nonetheless, in contrast, considering the TBARS results (Table 4), the catalysing effect of these metals on lipid oxidation in cooked patties during storage was weaker than the protective effect of BPS antioxidant compounds.

The effect of antioxidant treatment on other volatile compounds was considerably weaker than that described for lipid-derived compounds. Terpene compounds were scarcely affected by antioxidant treatment. Only camphene and caryophyllene contents showed significant differences. The former was higher in CON patties, and the latter was higher in ASC samples. Dimethyl sulphide was also affected, with higher concentrations in BPS than in ASC patties.

Changes in the lipid-derived compounds of the patty headspace resulting from ASC and BPS addition can influence patty flavour perception. The volatile compounds affected by antioxidant treatments generally have a relevant impact on cooked meat flavour [47], either directly in their original forms or as reactive intermediaries in the Maillard reaction [42]. The flavour of ASC patties may be milder, and that of BPS may be more intense. However, further sensory analysis is needed to confirm this hypothesis and to eventually evaluate the effect on consumers’ preferences.

## 4. Conclusions

The use of BPS improved the cooking yield of minced lean-pork batter to a similar extent to carrageenan, the conventional binder. In burger patties, the use of BPS seemed to prevent the discolouration of frozen-stored raw patties and diminish the lipid oxidation of cooked patties during refrigerated storage. The antioxidant effect of BPS at a concentration of 4.5 g BPS/kg may be comparable to the use of 0.5 g sodium ascorbate/kg (a conventional antioxidant at its typical amount in meat products). In contrast to ASC, the use of BPS increased the volatile compounds released by the patties due to heat treatment; however, this effect was reduced as the BPS amount was increased. The higher concentration of volatile compounds in the headspace of BPS burger patties would probably result in a change in flavour. Further research is needed to better understand the promoting effect of BPS on the release of volatile compounds and to assess whether this effect, together with the mouthfeel effect of BPS, could improve burger patty flavour perception. BPS is a potential functional ingredient for meat products that, in addition to improving their nutritional value, increases the water-holding capacity and oxidative stability of cooked meat products.

## Figures and Tables

**Table 1 foods-10-02173-t001:** Composition of banana pseudo-stem (BPS).

Component	
Moisture (%) ^1^	10.3
Crude protein (%) ^1^	6.8
Ether extract (%) ^1^	0.90
Ashes (%) ^1^	28.8
Fibre	
Total dietary fibre (%) ^1^	46.5
Neutral detergent fibre (%) ^2^	39.2
Acid detergent fibre (%) ^2^	27.4
Total extractable polyphenols (%) ^3^	2.32
Mineral content ^4^	
Potassium (%)	22.1
Phosphorous (%)	0.78
Calcium (%)	0.40
Magnesium (%)	0.31
Sodium (%)	0.11
Iron (mg/100 g)	37.5
Manganese (mg/100 g)	3.76
Zinc (mg/100 g)	3.20
Copper (mg/100 g)	0.95

^1^ AOAC [20]. ^2^ Van Soest, Robertson, and Lewis [21]. ^3^ Extracted in methanol and expressed as gallic acid equivalents; Folin and Ciocalteu [22]. ^4^ Osorio et al. [23].

**Table 2 foods-10-02173-t002:** Cooking loss and solids in the fluid lost, both expressed as percentages, and texture profile analysis results in minced lean-pork batter as a function of the amount and type of fibre-rich ingredient used (*n* = 5).

		L: 1.5 g/kg		M: 3.0 g/kg		H: 4.5 g/kg			
	CON	BPS	CAR	BPS	CAR	BPS	CAR	SEM	*p*-Value
pH	5.68 ^d^	5.70 ^c^	5.65 ^e^	5.73 ^b^	5.64 ^e^	5.76 ^a^	5.65 ^e^	0.011	***
Cooking loss	24.7 ^a^	22.6 ^ab^	21.8 ^bc^	19.0 ^c^	17.8 ^cd^	17.6 ^cd^	16.0 ^d^	1.26	***
Solids	2.66	2.43	2.62	2.33	2.34	2.11	2.21	0.200	NS
Hardness (N)	234 ^a^	220 ^ab^	213 ^bc^	219 ^bc^	218 ^bc^	205 ^c^	218 ^bc^	7.29	*
Elasticity	0.429	0.416	0.415	0.415	0.417	0.419	0.433	0.010	NS
Cohesiveness	0.736	0.786	0.788	0.785	0.741	0.764	0.753	0.027	NS
Chewiness (N)	73.9	72.6	69.1	72.0	67.2	65.8	70.8	4.45	NS

L, M, and H = 1.5 g, 3.0 g, and 4.5 g of fibre-rich ingredient per kg of batter, respectively; CON = control (no addition of fibre-rich ingredients); BPS = banana pseudo-stem; CAR = carrageenan; SEM = standard error of the mean; *p*-value: NS = not significant; * *p* < 0.05; *** *p* < 0.001. ^abcde^ Different superscripts in the same row indicate statistical differences by the Tukey test (*p* < 0.05).

**Table 3 foods-10-02173-t003:** Colour value and reflectance indexes of recently prepared (fresh) and frozen-stored ^#^ patties as a function of the type and amount of antioxidant used (*n* = 5).

	CON	ASC	LBPS	MBPS	HBPS	SEM	*p*-Value
Colour values							
*L**							
Fresh	47.8 ^a^	47.9 ^a^	45.9 ^b^	45.0 ^c^	44.1 ^d^	0.224	***
Frozen stored	45.8	45.9	44.5	43.5	42.3	1.073	NS
*a**							
Fresh	4.59 ^1^	4.64 ^1^	4.81 ^1^	4.89 ^1^	5.12 ^1^	0.181	NS
Frozen stored	2.74 ^2^	3.13 ^2^	3.28 ^2^	3.56 ^2^	3.93 ^2^	0.292	NS
*b**							
Fresh	11.8	11.8	12.1	12.2	12.4	0.290	NS
Frozen stored	10.3	10.4	11.0	11.3	11.6	0.383	NS
Chroma							
Fresh	12.7 ^1^	12.7 ^1^	13.0 ^1^	13.1 ^1^	13.5 ^1^	0.310	NS
Frozen stored	10.6 ^2^	10.9 ^2^	11.5 ^2^	11.9 ^2^	12.2 ^2^	0.389	NS
Hue angle							
Fresh	69.1 ^2^	68.7 ^2^	68.4 ^2^	68.1 ^2^	67.6 ^2^	0.715	NS
Frozen stored	75.1 ^1^	73.5 ^1^	73.3 ^1^	72.3 ^1^	71.2 ^1^	1.037	NS
Reflectance ratios ^&^							
R_λ630_/R_λ580_							
Fresh	1.88 ^a1^	1.91 ^a1^	1.70 ^b1^	1.59 ^bc1^	1.55 ^c1^	0.017	***
Frozen stored	1.35 ^2^	1.43 ^2^	1.34 ^2^	1.33 ^2^	1.32 ^2^	0.027	NS
Decrease ^$^	0.529 ^a^	0.477 ^a^	0.368 ^ab^	0.262 ^b^	0.223 ^b^	0.017	**
R_λ572_/R_λ525_							
Fresh	0.875 ^2^	0.869 ^2^	0.936 ^2^	0.985 ^2^	1.015	0.002	***
Frozen stored	0.998 ^1^	0.975 ^1^	1.029 ^1^	1.048 ^1^	1.065	0.010	NS
Increase ^$^	0.123 ^a^	0.105 ^a^	0.092 ^ab^	0.063 ^bc^	0.050 ^c^	0.002	***

^#^ Patties were stored at −18 °C for three months, wrapped in cling film, and thawed before colour measurement. CON = control (no addition of antioxidant ingredient); ASC = addition of 0.5 g sodium ascorbate per kg of patty; LBSE, MBSE, and HBSE = addition of 1.5 g, 3.0 g, and 4.5 g of banana pseudo-stem, respectively; SEM = standard error of the mean; *p*-value: NS = not significant; ** *p* < 0.01; *** *p* < 0.001. ^abcd^ Different superscripts in the same row indicate statistical differences by the Tukey test (*p* < 0.05). ^12^ Different superscripts in the same column for each characteristic indicate statistical differences by Student’s *t*-test (*p* < 0.05). ^&^ Reflectance ratios of selected wavelengths (λ) related to meat discolouration during storage [39] and increases or decreases in the ratios due to frozen storage. ^$^ Decrease or increase in the corresponding reflectance ratios due to frozen storage (calculated as the ratio in fresh patties minus the ratio in frozen patties).

**Table 4 foods-10-02173-t004:** Thiobarbituric acid reactive substances in recently cooked (not stored), refrigerated, and frozen stored patties (mg malondialdehyde/kg) as a function of the type and amount of antioxidant used (*n* = 5).

	CON	ASC	LBPS	MBPS	HBPS	SEM	*p*-Value
Not stored	0.60 ^a3^	0.14 ^e3^	0.44 ^b3^	0.32 ^c3^	0.24 ^d3^	0.022	***
Refrigerated stored ^&^	2.49 ^a1^	1.23 ^c1^	1.99 ^b1^	1.27 ^c1^	1.02 ^d1^	0.079	***
Frozen stored ^#^	1.19 ^a2^	0.51 ^c2^	1.31 ^a2^	1.18 ^a2^	0.82 ^b2^	0.067	***
SEM	0.075	0.037	0.054	0.054	0.076		
*p*-value	***	***	***	**	**		

^&^ After cooking, patties were stored at 4 °C for 48 h, wrapped in cling film, before TBARS measurement. ^#^ After cooking, patties were frozen and stored at −18 °C for three months, wrapped in cling film, and thawed before TBARS measurement. CON = control (no addition of antioxidant ingredient); ASC = addition of 0.5 g sodium ascorbate per kg of patty; LBSE, MBSE, and HBSE = addition of 1.5 g, 3.0 g, and 4.5 g of banana pseudo-stem, respectively; SEM = standard error of the mean; *p*-value: ** *p* < 0.01; *** *p* < 0.001. ^abcde^ Different superscripts in the same row indicate statistical differences by the Tukey test (*p* < 0.05). ^123^ Different superscripts in the same column indicate statistical differences by the Tukey test (*p* < 0.05).

**Table 5 foods-10-02173-t005:** Major ^&^ and volatile compounds (ng equivalent of hexanal per mL of headspace) in the headspace of cooked pork patties as a function of the type and amount of antioxidant used (*n* = 5).

	LRI	CON	ASC	LBPS	MBPS	HBPS	SEM	*p*-Value
Aldehydes and ketones		369.0 ^c^	215.0 ^d^	643.5 ^a^	542.2 ^b^	543.1 ^b^	35.30	***
Hexanal	807	237.7 ^c^	134.1 ^d^	424.9 ^a^	360.1 ^b^	342.3 ^b^	20.76	***
2-Heptanona	901	2.7 ^c^	1.0 ^d^	7.0 ^a^	4.7 ^b^	3.8 ^bc^	0.41	**
Heptanal	907	32.7 ^c^	20.9 ^d^	53.7 ^a^	46.1 ^b^	46.55 ^ab^	3.83	*
Octanal	1007	23.8 ^c^	12.5 ^d^	47.1 ^a^	35.5 ^b^	39.4 ^ab^	4.19	*
2-Octenal	1078	1.2	1.4	2.7	2.1	2.6	0.32	NS
Nonanal	1108	64.5 ^b^	41.8 ^c^	98.7 ^a^	84.8 ^a^	96.2 ^a^	6.05	**
Methyl-2-nonenal	1226	2.7 ^b^	1.9 ^c^	3.4 ^ab^	3.1 ^ab^	3.8 ^a^	0.23	*
2-Decenal	1287	0.9	0.4	2.3	2.6	4.2	0.46	NS
Alcohols		23.3	8.5	33.1	28.7	33.1	3.09	NS
Pentanol	770	13.5	4.8	19.4	15.9	19.0	1.72	NS
Octanol	1087	9.3	3.7	12.5	11.0	11.9	1.26	NS
Furans		161.7 ^c^	95.1 ^d^	250.7 ^a^	202.3 ^b^	204.0 ^b^	13.52	**
2-Butylfuran	893	3.2 ^a^	1.7 ^b^	4.0 ^a^	3.8 ^a^	3.0 ^a^	0.28	*
2-Pentyl	989	151.6 ^c^	89.3 ^d^	235.5 ^a^	188.2 ^bc^	190.6 ^b^	12.35	**
2-Hexylfuran	1091	2.6	1.5	3.9	3.8	3.8	0.39	NS
2-Heptylfuran	1203	1.9 ^b^	0.8 ^c^	2.7 ^a^	2.3 ^ab^	2.9 ^a^	0.25	*
Aliphatic hydrocarbons		18.6 ^b^	11.0 ^c^	24.2 ^a^	20.8 ^ab^	23.2 ^a^	1.73	*
Octane	800	11.4 ^b^	6.0 ^c^	15.1 ^a^	13.5 ^ab^	14.8 ^ab^	1.18	*
Terpene compounds		300.1	280.5	269.0	239.5	255.1	8.47	NS
α-Pinene	932	30.5	32.9	28.7	23.7	26.8	1.58	NS
Camphene	953	4.6 ^a^	3.5 ^b^	3.6 ^b^	3.2 ^b^	3.4 ^b^	0.16	*
β-Pinene	986	27.1	24.4	23.7	20.6	21.8	0.77	NS
Myrcene	991	7.0	7.7	6.3	5.6	6.3	0.26	NS
α-Phellandrene	993	3.7	7.1	2.9	3.7	3.3	1.06	NS
δ-Carene	1008	119.4	108.5	103.2	97.7	100.0	3.50	NS
p-Cymene	1026	12.7	8.2	10.6	10.1	11.1	0.52	NS
Limonene	1028	85.5	78.3	78.6	66.6	74.1	2.36	NS
β-Caryophyllene	1447	4.5 ^b^	6.0 ^a^	3.9 ^b^	3.5 ^b^	3.6 ^b^	0.22	**
Sulphur compounds		7.7 ^ab^	5.6 ^b^	10.3 ^a^	8.9 ^a^	8.7 ^a^	0.48	*
Dimethyl disulphide	755	7.5 ^ab^	5.6 ^b^	9.8 ^a^	8.4 ^a^	8.4 ^a^	0.45	*
Total sum of volatiles		883.9 ^c^	618.2 ^d^	1234.9 ^a^	1045.9 ^b^	1071.0 ^b^	54.14	**

^&^ Only compounds showing >4 ng equivalent of hexanal/mL of headspace in any of the treatments are included. LRI = linear retention index; CON = control (no addition of antioxidant ingredient); ASC = addition of 0.5 g sodium ascorbate per kg of patty; LBSE, MBSE, and HBSE = addition of 1.5 g, 3.0 g, and 4.5 g of banana pseudo-stem, respectively; SEM = standard error of the mean; *p*-value: * *p* < 0.05; ** *p* < 0.01; *** *p* < 0.001. ^abcd^ Different superscripts in the same row indicate statistical differences by the Tukey test (*p* < 0.05).

## Data Availability

The data presented in this study are available on request from the corresponding author.

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
