# Peer review of "Banana Pseudo-Stem Increases the Water-Holding Capacity of Minced Pork Batter and the Oxidative Stability of Pork Patties"

_foods, 2021, doi:10.3390/foods10092173_

Round 1
Reviewer 1 Report
The manuscript presents the potential of banana pseudo-stem (BPS) as a functional ingredient in meat derivate, a pork patty, with a particular focus on its filler and antioxidant effects. This manuscript presents relevant information however, some sections of the introduction, materials and methods and conclusion can be improved. For this reason, I considered that this manuscript needs some changes.
Introduction - Please specify exactly what is the scientific novelty of this study.
lines 67-69 Please modify the topic of work. In experiment 1 it was checked whether BPS can be used as a substitute for CAR in cooked meat products, in experiment 2 the possibility of using BPS as an ascorbate substitute was determined.
line 108 Why was it decided to measure the parameter only after 2 days?Why the model product was not stored longer (minimum 7 days)? Has the microbiological purity of the BPS/meat patties been tested?
Experiment plan:
It is difficult for the reader to understand what technological processing (heat treatment) the pates have been subjected to. Why in experiment 1 it was decided to cooked in water and in experiment 2 - cooked in convection oven?
Please change the titles and subsection numbers so that they are not misleading 2.2., 2.3., 2.4. x 3).
lines 151-152 You mixed 3 kg of meat with additives by hand? Does this procedure ensure even distribution of the additives?
line 235 Please do not forget that you are also analyzing the effect of BPS additive on the color of the product. The additive itself changes the color of the product (darkening).
Conclusion:
lines 384-385 A agree, hovewer not in all research variants (see frozen stored, table 4).
line 389 Has a sensory, organoleptic or consumer analysis of the product been carried out?
Author Response
Many thanks for the suggestions and comments, which we have addresses point by point as follows:
The manuscript presents the potential of banana pseudo-stem (BPS) as a functional ingredient in meat derivate, a pork patty, with a particular focus on its filler and antioxidant effects. This manuscript presents relevant information however, some sections of the introduction, materials and methods and conclusion can be improved. For this reason, I considered that this manuscript needs some changes.
Introduction - Please specify exactly what is the scientific novelty of this study.
Answer: The novelty of the study has been clearly stated at the end of the introduction section
lines 67-69 Please modify the topic of work. In experiment 1 it was checked whether BPS can be used as a substitute for CAR in cooked meat products, in experiment 2 the possibility of using BPS as an ascorbate substitute was determined.
Answer: The sentence has been rewritten accordingly, i.e. providing more specific information.
line 108 Why was it decided to measure the parameter only after 2 days? Why the model product was not stored longer (minimum 7 days)?
Answer: We think that one-three days are a usual storage length for aerobically stored cooked patties at home in the fridge, or at food services or pre-cooked food facilities. Moreover, our experience in the lab is that cooked meat and cooked fresh-meat products such as burgers show a sharp increase in their TBARS values during the first one-two days of storage, and then, the increase is weaker increases and tends to stabilize. It can also be seen in the study by Bañón et al. Meat Science 77 (2007) 626–633 https://reader.elsevier.com/reader/sd/pii/S0309174007001738?token=3FA20696337587902763040527560916F8975ACF7DC82C4CA0A953179700BE62FD290BC5ABAE129DAD95ADAB4A7D5833&originRegion=eu-west-1&originCreation=20210830090752
Has the microbiological purity of the BPS/meat patties been tested?
Answer: The drying process with hot air (minimum 65 ºC) would control the microbial load. The provider told us that the microbial load is around Log 5-6 UFC/g. We have not performed microbial analysis.
Experiment plan:
It is difficult for the reader to understand what technological processing (heat treatment) the pates have been subjected to. Why in experiment 1 it was decided to cooked in water and in experiment 2 - cooked in convection oven?
Answer: Thanks for your comment. We have tried to clarify the point and make it easy to understand. We have started by modifying the title to differentiate the two experiments (the performance of BPS on the water holding capacity of a pork batter and the performance of BPS on the oxidative stability of a meat product). Then, we have reworded the Experiment plant to make clear that each experiment was independently enough to deserve a different heat treatment.
Please change the titles and subsection numbers so that they are not misleading 2.2., 2.3., 2.4. x 3).
Answer: Thank you. The numbers of the subsections have been changed for correctness.
lines 151-152 You mixed 3 kg of meat with additives by hand? Does this procedure ensure even distribution of the additives?
Answer: We have added the word vigorously before mixed. We payed especial attention to the mixing step the batter was vigorously kneaded with hand cycles of approximately 1 per sec.
line 235 Please do not forget that you are also analyzing the effect of BPS additive on the color of the product. The additive itself changes the color of the product (darkening).
Answer: The subtitle has been reworded to include colour determination and more emphasis on the darkening effect of BPS in the fresh patties has been given in the text.
Conclusion:
lines 384-385 A agree, hovewer not in all research variants (see frozen stored, table 4).
Answer: Thank you. The sentence has been rewritten to fulfil the request.
line 389 Has a sensory, organoleptic or consumer analysis of the product been carried out?
Answer: The sentence has been rewritten for higher accuracy. And the answer to the question is that it was made a preliminary consumer analysis with cooked pork batters to assess whether a level of 4.5 g BPS/kg was accepted/rejected by consumers and found that was accepted (2.2. Experiment plan). However, no analysis was carried out in patty burgers.
Reviewer 2 Report
The article entitled “Banana pseudo-stem increases the water holding capacity and oxidative stability in pork patties” gives interesting insights on the feasibility of using banana’s by-products in meat industries to replace some conventional ingredients. The study is strongly focused on assessing the effects of Banana pseudo-stem on physical properties and oxidative parameters of meat butters and patties. The experimental design is sound, and the number of samples and replicate are appropriate to give reliable information on the effects of this ingredient on the studied traits. However, some minor amendments are needed.
Please find below some general and specific comments:
Abstract:
Abstract can be improved, at first glance the presentation of your study and of the results is a bit fragmented.
Moreover:
L15: Banana pseudo-stem (BPS), a potential functional ingredient rich in fibre and polyphenols, was used in lean pork batters and patties at levels of 0.0 (control, CON), 1.5, 3.0, and 4.5 g/kg -> I suggest to also add “BPS binder and antioxidant properties were tested in comparison with conventional carrageenan and sodium ascorbate, respectively” or something similar to outline that beside the CON group you also tested conventional ingredients, which is currently missing from the abstract.
L16: I would replace metmyoglobin with discoloration since you did not assess metmyoglobin directly, but this was part of you interpretation of color data. In the abstract I would suggest of focus on the analysis you have carried out.
Introduction
L36: there is a range
L57: performed/provided instead of found
Material and methods
L80: I would add the abbreviation (BPS) at the end of Table 1 caption
L87: Please delete “experiments one and two”, is redundant
L87-90: I would break this sentence, for instance: The first experiment evaluated the effect of the use of BPS as a filler in lean-pork burger patty batters on technological quality traits. BPS was compared to a control (CON) patty batter (with no filler), and patty batters with carrageenan (CAR).
L171: The remaining halves were used
L203-2004: The outputs of this part of statistical analysis were not reported in the manuscript
Results and discussion
L207: 4.5 g/kg
L209: I would suggest expanding this part on pH discussion reporting also that pH increases accordingly with the increase of BPS amount, whereas CAR seems not to affect pH which remains unchanged regardless of CAR content
L224-225: I suggest breaking the sentence: “…quality traits studied. Both ingredients improved the cooking yield…”
L249: “in both CON and ASC patties by significant decreases”
L251: Please delete reference 39, I think it is misplaced in this row
L260: I suggest rephrasing “These changes could be the result of a direct effect of BPS on the light reflectance of patties rather than a particular effect of BPS on the proportions of the three myoglobin chemical forms”
L263: Table 3. It is not very clear to what the Delta is referred. I would suggest adding an apex and a footnote to the table where you report how it is calculated, like you have written in the text L278-279”. Moreover, the symbol of the first Delta is wrong.
L281: I think that reference 39 is again in the wrong place.
L284-285: Why did you not reported this data in the table? If you have them, it would be interesting to display them, also considering the novelty of you study.
L292: Table 4. I think it could be slightly improved by removing the apexes from the caption and putting them on the respective rows. Moreover, I think the footnote “#” is wrong. You reported that are raw patties frozen and stored, but in M&M section L163-165, you also report that patties were all cooked before being divided for storage treatment and lipid oxidation assessment.
L317: Please delete “with which”, it is not needed, and the sentence could become more flowing
L329: Table 5 caption, please replace lamb with pork
The discussion on results displayed in Table 5 is in some points a bit chaotic and it makes difficult to follow your thinking. Especially from L349 to L369. Please check the English and I would suggest making shorter sentences.
L350-354 and L362-369: Please carefully rephrase these sentences, they are very unclear
L375: Please replace ascorbate with ASC
L376 Please replace these with the
L381: consumer’s preference
In general, in the manuscript there are several sentences too long, which make difficult to follow you argumentation. I would suggest reading it again and cut some of them. For example, L33-36.
Author Response
Many thanks for the suggestions and comments, which we have addresses point by point as follows:
Abstract can be improved, at first glance the presentation of your study and of the results is a bit fragmented.
Answer: The abstract has been rewritten in an attempt to make it more coherent and give it flow
Moreover:
L15: Banana pseudo-stem (BPS), a potential functional ingredient rich in fibre and polyphenols, was used in lean pork batters and patties at levels of 0.0 (control, CON), 1.5, 3.0, and 4.5 g/kg -> I suggest to also add “BPS binder and antioxidant properties were tested in comparison with conventional carrageenan and sodium ascorbate, respectively” or something similar to outline that beside the CON group you also tested conventional ingredients, which is currently missing from the abstract.
Answer: The suggestion has been attended
L16: I would replace metmyoglobin with discoloration since you did not assess metmyoglobin directly, but this was part of you interpretation of color data. In the abstract I would suggest of focus on the analysis you have carried out.
Answer: Replaced accordingly
Introduction
L36: there is a range
Answer: Corrected
L57: performed/provided instead of found
Answer: Replaced with provided
Material and methods
L80: I would add the abbreviation (BPS) at the end of Table 1 caption
Answer: BPS has been added accordingly
L87: Please delete “experiments one and two”, is redundant
Answer: Deleted
L87-90: I would break this sentence, for instance: The first experiment evaluated the effect of the use of BPS as a filler in lean-pork burger patty batters on technological quality traits. BPS was compared to a control (CON) patty batter (with no filler), and patty batters with carrageenan (CAR).
Answer: Done.
L171: The remaining halves were used
Answer: Changed
L203-204: The outputs of this part of statistical analysis were not reported in the manuscript
Answer: The Pearson correlation analysis was removed from the statistical analysis section
Results and discussion
L207: 4.5 g/kg
Answer: Thank you; it was amended
L209: I would suggest expanding this part on pH discussion reporting also that pH increases accordingly with the increase of BPS amount, whereas CAR seems not to affect pH which remains unchanged regardless of CAR content
Answer: The suggestion has been considered
L224-225: I suggest breaking the sentence: “…quality traits studied. Both ingredients improved the cooking yield…”
Answer: Done
L249: “in both CON and ASC patties by significant decreases”
Answer: Modified
L251: Please delete reference 39, I think it is misplaced in this row
Answer: The URL of the reference was misleading. We have changed the URL of the reference to the correct one.
L260: I suggest rephrasing “These changes could be the result of a direct effect of BPS on the light reflectance of patties rather than a particular effect of BPS on the proportions of the three myoglobin chemical forms”
Answer: Thank you. It was done
L263: Table 3. It is not very clear to what the Delta is referred. I would suggest adding an apex and a footnote to the table where you report how it is calculated, like you have written in the text L278-279”. Moreover, the symbol of the first Delta is wrong.
Answer: It has been changed to improve clarity
L281: I think that reference 39 is again in the wrong place.
Answer: See comment above.
L284-285: Why did you not reported this data in the table? If you have them, it would be interesting to display them, also considering the novelty of you study.
Answer: The data has been included in the Table
L292: Table 4. I think it could be slightly improved by removing the apexes from the caption and putting them on the respective rows. Moreover, I think the footnote “#” is wrong. You reported that are raw patties frozen and stored, but in M&M section L163-165, you also report that patties were all cooked before being divided for storage treatment and lipid oxidation assessment.
Answer: Thank you. Both of the remarks have been ammended.
L317: Please delete “with which”, it is not needed, and the sentence could become more flowing
Answer: Deleted
L329: Table 5 caption, please replace lamb with pork
Answer: Replaced
The discussion on results displayed in Table 5 is in some points a bit chaotic and it makes difficult to follow your thinking. Especially from L349 to L369. Please check the English and I would suggest making shorter sentences. L350-354 and L362-369: Please carefully rephrase these sentences, they are very unclear
Answer: The text in these lines has been revised and rephrase accordingly
L375: Please replace ascorbate with ASC
Answer: Replaced
L376 Please replace these with the
Answer: Replaced
L381: consumer’s preference
Answer: Changed
In general, in the manuscript there are several sentences too long, which make difficult to follow you argumentation. I would suggest reading it again and cut some of them. For example, L33-36.
Answer: Done. The manuscript has been sent to a professional editing service.